# Exogenous Oestrogen Impacts Cell Fate Decision in the Developing Gonads: A Potential Cause of Declining Human Reproductive Health

**DOI:** 10.3390/ijms21218377

**Published:** 2020-11-08

**Authors:** Melanie K. Stewart, Deidre M. Mattiske, Andrew J. Pask

**Affiliations:** School of BioSciences, The University of Melbourne, Melbourne, VIC 3010, Australia; deidre.mattiske@unimelb.edu.au (D.M.M.); a.pask@unimelb.edu.au (A.J.P.)

**Keywords:** gonad, oestrogen, endocrine disrupting chemicals, differences of sexual development, fertility, SOX9

## Abstract

The increasing incidence of testicular dysgenesis syndrome-related conditions and overall decline in human fertility has been linked to the prevalence of oestrogenic endocrine disrupting chemicals (EDCs) in the environment. Ectopic activation of oestrogen signalling by EDCs in the gonad can impact testis and ovary function and development. Oestrogen is the critical driver of ovarian differentiation in non-mammalian vertebrates, and in its absence a testis will form. In contrast, oestrogen is not required for mammalian ovarian differentiation, but it is essential for its maintenance, illustrating it is necessary for reinforcing ovarian fate. Interestingly, exposure of the bi-potential gonad to exogenous oestrogen can cause XY sex reversal in marsupials and this is mediated by the cytoplasmic retention of the testis-determining factor SOX9 (sex-determining region Y box transcription factor 9). Oestrogen can similarly suppress SOX9 and activate ovarian genes in both humans and mice, demonstrating it plays an essential role in all mammals in mediating gonad somatic cell fate. Here, we review the molecular control of gonad differentiation and explore the mechanisms through which exogenous oestrogen can influence somatic cell fate to disrupt gonad development and function. Understanding these mechanisms is essential for defining the effects of oestrogenic EDCs on the developing gonads and ultimately their impacts on human reproductive health.

## 1. Introduction

Gonadal sex determination is the process through which the bi-potential gonad differentiates into either an ovary or testis. This leads to the development of corresponding female or male secondary sex characteristics and has profound effects on the subsequent physiology and behaviour of the organism. The bi-potential gonad is comprised of the machinery required to follow one of two fates—ovary or testis—and this is under the control of well-defined molecular pathways [1,2]. The somatic cells of the gonad are integral for influencing the overall fate of the gonad such that the differentiation of these cells is the critical first step in the development of the reproductive tract. Mouse models have demonstrated that these somatic cells display plasticity, where loss or gain of key gonadal genes can drive granulosa (ovary) or Sertoli (testis) cell differentiation, independent of chromosomal sex [3,4,5,6,7,8,9,10]. Interestingly, oestrogen is able to influence these pathways in XY mammalian Sertoli cells to promote granulosa-like cell fate [11,12]. Even brief disruptions to testicular signalling pathways can impact Sertoli cell patterning, disrupting the development and function of the testis. This is of particular concern given our increasing exposures to endocrine disrupting chemicals (EDCs) that can interact with native oestrogen receptors and the decline in human reproductive health over recent decades [13].

## 2. The Impact of Oestrogenic Endocrine Disrupting Chemicals on Reproductive Health

Over the last 50 years, reproductive health has rapidly declined as a result of both increasing infertility and occurrence of reproductive birth defects. In males, a 50% decrease in sperm counts has been observed [14], alongside increasing rates of testicular cancer [15] and abnormalities in the development of the reproductive tract known as differences of sexual development (DSDs) [16,17]. DSDs are some of the most common birth defects in humans, affecting gonadal and anatomic sex development and occurring in up to 1:200 live births [18]. Testicular dysgenesis syndrome (TDS) comprises some of these conditions, including hypospadias, cryptorchidism, testicular cancer, and poor semen quality [19]. TDS is thought to arise from disruptions to the development and functioning of the testis during early fetal life, leading to compromised differentiation of the reproductive tract [20]. Hypospadias is one of the most frequently occurring birth defects in males, affecting 1:125 live male births in Australia [21]; however, only 30% of hypospadias cases can be attributed to genetic factors [22], suggesting a substantial environmental component is involved in the development of this condition. Furthermore, the increasing prevalence of TDS-related conditions has occurred too rapidly to be caused by genetic mutation alone and instead has been linked to our continued exposure to endocrine disrupting chemicals (EDCs) [23,24,25,26,27,28]. 

EDCs are defined as “an exogenous substance or mixture that alters function(s) of the endocrine system and consequently causes adverse health effects in an intact organism, or its progeny, or (sub) populations” [29]. EDCs can target specific hormonal pathways by interacting directly with receptors; for instance, some EDCs are capable of binding to native oestrogen receptors (ERs) to trigger the ectopic activation of oestrogen-responsive signalling pathways [30]. These oestrogenic EDCs are some of the most pervasive in our environment and include compounds, such as bisphenol A (BPA; a plasticiser), 17α-ethynylestradiol (a component of the contraceptive pill), oestrogenic phthalates (DEHP, DBP, DBP [31,32,33]; plasticisers and present in cosmetics), and genistein (a phytoestrogen naturally occurring in soy and subterranean clover; Figure 1). Aberrant activation of oestrogen signalling is detrimental to development as the correct levels of oestrogen are imperative for sexual differentiation of both the male and female reproductive tract. Several studies have demonstrated that reproductive development requires a delicate balance of androgens and oestrogens [34,35,36]; furthermore, the embryonic mammalian gonad expresses oestrogen receptors throughout development [37,38,39] and is therefore a direct target of oestrogenic chemicals. However, the predominant oestrogen receptor subtype appears to differ between mammalian species [40].

The correct patterning of the gonad is crucial for establishing the cells that produce androgens and oestrogens and contribute to the differentiation of the urogenital tract. Studies in humans and mice have demonstrated that EDCs can interfere with gonad function and subsequently the differentiation of the male reproductive tract (Figure 1). In mice, exposure to the oestrogenic endocrine disruptor diethylstilbestrol (DES) in utero leads to increased rates of hypospadias and reduced anogenital distance [41,42], a marker of androgen output during development [43,44]. Such results have also been confirmed in vitro, where DES causes reduced testosterone output in mouse and rat gonad cultures [45] and BPA impairs testosterone production in human fetal testis culture [45,46]. Reduced synthesis of testosterone indicates impaired testis function, which can lead to disruption of the overall patterning and differentiation of the male reproductive tract. 

Associations between EDCs and TDS in humans are more difficult to elucidate given the lack of controlled conditions, but many studies have demonstrated a link between exposure to EDCs, such as genistein and BPA, in utero and the development of hypospadias and TDS-related conditions [25,28,47,48,49]. The detrimental effects of EDCs are also of concern in adulthood, where a high level of BPA in urine is associated with reduced sperm counts and motility [50,51] and elevated exogenous oestrogen levels during adulthood negatively affect testis function in humans [52,53]. Together, these data demonstrate the ability of EDCs to target the testis in mammals, causing decreased testis function and subsequent disruption of male reproductive tract differentiation and fertility (Figure 1). 

Females can similarly be impacted by excess oestrogen signalling, as the early development of the female urogenital tract occurs in the absence of any hormones [54] such that exposure to oestrogen at this time is also ectopic, leading to the development of conditions associated with ovarian dysgenesis syndrome [55]. The best characterised case of exogenous oestrogen signalling impacting female reproductive development is the daughters of DES treated women. DES was prescribed to pregnant women between 1938 and 1975 to prevent miscarriage or premature birth [56,57]. 5–10 million women were prescribed DES in the U.S. alone [58], and the drug was also widely used throughout Europe, Australia and the UK. Not only was DES ineffective in preventing miscarriage, but it caused an increased incidence of reproductive tract cancers, infertility and recurrent miscarriage in the daughters of women exposed to DES [59,60]. Thus, it is clear that exogenous oestrogen signalling is also detrimental to female reproductive health.

Aberrant oestrogen signalling during critical periods of development and even in adulthood can impact the function of the ovary. The age of onset of puberty in girls has decreased in the U.S., Denmark, India, and China [61,62,63,64] and is thought to similarly be linked to increasing oestrogenic EDC exposure. Sex steroids play a crucial role in pubertal timing, and disruption of this timing—such as via exposure to EDCs—can have long-term reproductive consequences [65]. 

Exposure to oestrogenic EDCs has also been linked to conditions caused by compromised ovarian function and depletion of ovarian reserve (Figure 1), such as polycystic ovary syndrome (PCOS) [66] and primary ovarian insufficiency (POI) [67,68]. PCOS affects between 15–20% of women of reproductive age and is the most commonly occurring endocrine disorder in women [66,69]. PCOS is characterised by hyperandrogenism, ovulatory dysfunction, and polycystic ovaries, alongside an increased risk of diabetes and cardiovascular disease [69]. POI is less widespread, with a global prevalence estimated to be 3.7%—a rate that has increased in recent years [70]—and is defined as cessation of menstruation prior to the typical age of menopause, contributing not only to fertility difficulties, but also an increased likelihood of cardiovascular disease, osteoporosis and depression [71,72]. Both PCOS and POI are characterised by a loss of oestrogen signalling and hyperandrogenism. Indeed, continued oestrogen signalling is essential for the maintenance of the ovary in mammals [73]; however, studies in rodents and cell lines have suggested that aberrant oestrogen signalling either during development or in adulthood can cause a reduction in the oestrogenic output of the ovary [67].

Numerous studies have demonstrated the ability of BPA and genistein to reduce the steroidogenic output of the ovary and impact folliculogenesis, raising concern about the harm of these chemicals on ovarian function [67,74]. High BPA blood levels are associated with PCOS in women [75] and exposure to BPA during development leads to formation of PCOS-like phenotypes during adult life in rats [76] and mice [77]. Similarly, exposure to either genistein or a mixture of oestrogenic and anti-androgenic EDCs can cause a reduction in follicular reserve and POI-like phenotypes in rats [78,79]. It is hypothesised that the development of these phenotypes is due to reduced oestrogen output and compromised ovarian function; indeed, follicles or granulosa cells cultured in the presence of BPA show a decrease in oestrogen production [80,81] and exposure to genistein decreases expression of critical steroidogenic pathways in human granulosa cells [82]. Overall, these results suggest a disruption to the key pathways involved in maintaining oestrogenic output and therefore ovarian identity. 

The impact of EDCs on reproductive health is concerning, particularly their ability to affect the development of the gonad and urogenital tract during fetal life, contributing to the rise in prevalence of DSDs. Their impact on reproductive health after birth and into adulthood is of further concern, where exposure to EDCs has been linked to premature puberty, PCOS and POI in females, and reduced sperm counts in men, together contributing to an overall decline in fertility. These issues primarily stem from the ability of oestrogenic chemicals to target the testis and ovary. The gonad harbours and nurtures the germ cells that will go on to form sperm and oocytes, and eventually the next generation. Additionally, the gonad synthesises the majority of sex hormones in males and females, which are essential for directing sexual differentiation and maintaining reproductive function. Examining the effect of oestrogen on development and maintenance of gonad fate (ovary or testis) and the molecular pathways that drive this process is critical to understanding how EDCs may target this system. 

## 3. The Function of Oestrogen in the Mammalian Gonad

Oestrogen has a critical role in mediating ovarian differentiation in non-mammalian vertebrates, regardless of the sex determining mechanism. An increase in oestrogen—through changes to endogenous or exogenous oestrogen levels—can consistently promote male-to-female sex reversal, demonstrating the plasticity of gonadal sex and ability of oestrogen to promote ovarian fate [83,84,85,86]. In contrast, the role of oestrogen in early gonadal development and its ability to promote differentiation is less clear in mammals. 

Exposure to exogenous oestrogen prior to gonad differentiation can cause sex reversal in two marsupials, the opossum [87] and tammar wallaby [88], despite their clear genetic sex determination system. This suggests that, similar to non-mammalian vertebrates, oestrogen can override the genetic predisposition of the gonad to become a testis. At present, the effect of exogenous oestrogen on other mammalian species is less clear, but there is a known role for the hormone in maintaining ovarian fate. Oestrogen is also essential for ovarian differentiation in goats [89], sheep [90,91], and cows [92], where aromatase promotes the synthesis of oestrogen from testosterone in the fetal ovary. 

In mice, the presence of oestrogen is not essential to induce the bi-potential gonad to actively differentiate into an ovary, but it is still necessary for the maintenance of somatic cells. Mice deficient for *Cyp19* (encodes aromatase) undergo normal early ovarian differentiation, illustrating that oestrogen is not required for initial development [3]. However, shortly after birth, the germ cells of these mice are lost and the gonad shows testis-like morphology, where the somatic cells change fate from granulosa (ovarian) to Sertoli (testis). Administration of oestrogen rescues this phenotype, demonstrating that the hormone can trigger cell fate change and is necessary for ongoing maintenance and function of granulosa cells [73]. Further demonstrating this requirement of oestrogen for ovarian maintenance, mice lacking oestrogen receptor α (ERαKO) have normal ovarian development until adulthood, when the ovary does not successfully complete folliculogenesis [93]. These mice still have some oestrogen signalling as they express ERβ, but these findings demonstrate the requirement of ERα for normal ovarian function.

In general, the role of oestrogen in early eutherian gonad development is downplayed given the presence of a strong genetic sex determination (GSD) system and the fact that the process of sex determination occurs in utero, where there could be exposure to maternal oestrogens. Given this, it has been assumed that oestrogen would have no impact on gonad differentiation and that the developing gonad would be resistant to the influence of any maternal oestrogens [94]. Despite this, ERs are widely expressed in the indifferent gonad of all mammals [37,38,39], making them susceptible to exposure to endocrine disruptors that can interact with ERs. Indeed, oestrogenic EDCs can cross the placenta and increase the typical levels of oestrogen in the uterine environment [95], bypassing any resistance provided by the placenta. Furthermore, the link between increasing oestrogenic EDCs and infertility and DSDs suggests that the gonad is a target of exogenous oestrogen.

While the precise role for oestrogen in directing early ovarian differentiation in mammals appears to be variable across species, it plays a highly conserved role in ovarian and granulosa cell maintenance. To further understand the function of oestrogen in regulating somatic cell fate, it is essential to understand the core pathways critical for mammalian gonad differentiation and examine where oestrogen can potentially influence this system. 

## 4. Molecular Control of Gonad Differentiation

Gonad development begins with the initial emergence of the bi-potential gonad, an indifferent structure that can form either an ovary or testis [2,96]. At embryonic (E) day 10.5 in mice (equivalent to the 6th week of gestation in humans), the bi-potential gonad emerges on the mesonephros, a process under the control of *Wt1*, *Sf1*, *Cbx2*, *Lhx9*, and *Emx2* [97]. Within the indifferent gonad are the supporting somatic cells, which can form either a testis-specific (Sertoli) or ovary-specific (granulosa) cell. 

### 4.1. Testis Development

Sertoli cells are the first cell type to differentiate in the male gonad and are considered the orchestrators of subsequent testis development [98]. Following the formation of a testis, Sertoli cells are involved in supporting steroidogenesis, spermatogenesis and maintenance of testis identity [99]. A minimum number of Sertoli cells is required for the development of a testis to continue [100], and because of this essential threshold of Sertoli cell number, the recruitment of Sertoli cells is an important process to ensure that testis development occurs correctly. Sertoli cell determination is marked by the temporally controlled expression of the Y chromosome gene sex-determining region Y (*Sry*) at E11.5 [101]. *Sry* is the molecular switch required for testis formation [102], and both the correct timing [103] and level [104] of its expression are necessary for testis development to occur. Indeed, the initial Sertoli cell recruitment and subsequent maintenance of the required Sertoli cell number is supported by expression of key testis factors downstream of *Sry.*


Once levels of *Sry* reach a critical threshold at E11.5, SRY-box transcription factor 9 (*Sox9*) transcription is initiated (Figure 2). Prior to this at E10.5, SOX9 is present in the cytoplasm of XX and XY indifferent gonad somatic cells [105]. Upon expression of *Sry* in XY mice embryos, SOX9 translocates to the nucleus; however, in the absence of *Sry*, the cytoplasmic pool of SOX9 dissipates [105]. SOX9 shows the same localisation pattern in humans [106] and this sex-specific regulation of SOX9 is the key trigger for testis differentiation in both species. Indeed, ectopic expression of *Sox9* in the indifferent XX mouse gonad is able to trigger testis differentiation [5,6], and the absence of *Sox9* in XY mice leads to formation of an ovary [7,8]. *Sox9* activity is sufficient to trigger all downstream testis development, even in the absence of *Sry* [107]. Furthermore, heterozygous mutations for SOX9 in humans can lead to XY sex reversal [108,109]. Consequently, SOX9 is considered a critical testis-determining gene and major emphasis has been placed on understanding its regulation and downstream role as a transcription factor.

The necessity of SOX9 to direct testis development relies on its ability to initiate transcription of downstream targets that further support testis formation and function. These downstream targets include fibroblast growth factor 9 (*FGF9*), prostaglandin D synthase (*PTGDS*), and anti-Müllerian hormone (*AMH;*
Figure 2). FGF9 is a secreted signalling molecule, and, during embryonic mouse development, *Fgf9* shows a sex-specific pattern of expression [110] before becoming restricted to XY gonads [111]. *Fgf9* forms a feed-forward positive loop with *Sox9* and suppresses the ovarian gene *Wnt4* [112] to promote testis formation. *Fgf9* null mice exhibit XY sex reversal in some, but not all, genetic backgrounds [113] and it has been hypothesised that this sex reversal is due to reduced proliferation rate and differentiation of pre-Sertoli cells [111]. These results demonstrate the role for FGF9 in recruiting Sertoli cells to the threshold required for formation of a testis, the failure of which results in sex reversal in mice [100]. Interestingly, a mutation in *FGFR2* (which encodes the FGF9 receptor) has been reported in an XY gonadal dysgenesis patient, suggesting that FGF9 signalling is also important for human testis development [114].

Similar to FGF9, *Ptgds* forms a feed-forward loop with SOX9 [115,116]. *Ptgds* produces PGD2, a paracrine factor secreted by Sertoli cells that promotes their differentiation and maintenance. PGD2 has also been implicated in the ability of XY somatic cells to recruit XX somatic cells to express *Sox9* when cultured together in vitro [115]. This demonstrates that, like FGF9, PGD2 is required for maintaining the threshold of Sertoli cells required for testis development. *Ptgds* is expressed in a male-specific manner in embryonic mouse gonads from E11.5 to E14.5 [116,117], and loss of *Ptgds* in XY mice leads to reduced *Sox9* transcription and delayed testis cord formation [118]. Interestingly, culture of XX gonads in the presence of PGD2 can induce testicular cord formation and expression of testis-specific genes [117], further illustrating it has a strong testis-promoting function.

*Sox9* also initiates expression of *Amh* and works with steroidogenic factor 1 (*Sf1*) to maintain production of the hormone in Sertoli cells [119,120]. AMH is responsible for the regression of the Müllerian ducts, a structure that, when present, is a key characteristic of female development [121]. Transgenic female mice chronically expressing *Amh* develop abnormally, with complete absence of a uterus or oviducts and disrupted ovarian function [122]. *Amh* is therefore critical for establishing normal sexual differentiation and promoting male development. Together, the expression of *SOX9, FGF9*, *PTGDS*, and *AMH* work to establish the specification and proliferation of Sertoli cells, contributing to the initial differentiation of the testis and ultimately a functioning male reproductive system. 

### 4.2. Ovarian Development

In XX gonads, ovary-specific genes are expressed following the disappearance of cytoplasmic SOX9 [105]. This includes R-spondin 1 (*Rspo1*) and the Wnt/β-catenin pathway, which become specific to granulosa cells at E12.5 [123,124]. *RSPO1* has more recently been considered to be the critical female-determining gene. The requirement for *RSPO1* in ovarian determination was initially discovered by linking human *RSPO1* mutations to XX gonadal dysgenesis [124]. Similarly, *Rspo1* null mutant XX mice exhibit masculinisation of the gonad and some expression of *Sox9* [10]. *Rspo1* can stabilise β-catenin (encoded by *Ctnnb1*) [125], leading to activation of the *Wnt4*/β-catenin pathway that is essential to drive ovarian differentiation in early development [10] (Figure 2). β-catenin has similar ovary-promoting effects and, when stabilised, can enter the nucleus and act on target genes by increasing expression of *Lef1* in a female-specific pattern [10]. Ectopic stabilisation of β-catenin in XY gonads can cause male-to-female sex reversal in mice [4], demonstrating it can promote ovarian differentiation in the presence of SOX9.

WNT/β-catenin signalling activates numerous downstream targets that are essential for ovarian development; in particular, increased β-catenin activity can induce expression of *FoxL2* [126]. *FoxL2* is expressed in XX gonads from E12.5 and is necessary for the specification and maintenance of granulosa cell fate [127]. Loss of *FoxL2* has no impact on the early development of the ovary, suggesting it is not the critical ovary-determining gene; however, its ablation in adult mouse ovaries leads to transdifferentiation of granulosa cells to a Sertoli cell phenotype and upregulation of *Sox9* [9], demonstrating it has a strong antagonistic relationship with *Sox9* and is required for maintaining granulosa cell fate. Furthermore, overactivation of β-catenin in mice testes during development leads to increased expression of *FoxL2* and drives transformation of Sertoli cells to granulosa-like cells [128], while ectopic expression of *FoxL2* in embryonic mouse testes represses Sertoli cell differentiation and causes partial male-to-female sex reversal [129]. 

FOXL2 appears to have a role in ovarian maintenance in humans, as mutations in the gene cause premature ovarian insufficiency [130]. This role of FOXL2 in granulosa cell maintenance is similar to that of oestrogen [73]. Interestingly, the absence of *FOXL2* in goats causes XX sex reversal [131], suggesting there exists a more critical role for the gene in ovarian determination in some mammals. Oestrogen is also required for the early differentiation of the ovary in goats [89], further suggesting a relationship between FOXL2 and oestrogen in mammals. FOXL2 is important for ERβ signalling in mouse ovary [132], and it has been established that ERs have a close relationship with other forkhead box transcription factors, as well [133,134]. In particular, ER transcriptional activity in breast cancer is dependent on its binding to forkhead box A1 (FOXA1) [135]. Thus, it is likely there exists a similar interaction between ERs and FOXL2. Together, these ovarian genes establish the identity of granulosa cells and their continued maintenance, working to suppress the male developmental pathway, while promoting ovarian differentiation and function.

### 4.3. Antagonism between Pro-Testis and Pro-Ovarian Factors Drives Sex Determination

Numerous pro-ovary and pro-testis factors in the gonad determination pathway exhibit opposing effects (Figure 2). This pathway antagonism has led to the establishment of a ‘push-and-pull’ model, wherein the somatic cells of the gonad are plastic in nature and their fate is dependent on the level of pro-ovary or pro-testis factors. Indeed, the ability of oestrogen to impact somatic cell fate relies on this plasticity and takes advantage of the push and pull between gonad developmental pathways. 

*Wnt4* has an antagonistic relationship with the testis-specific gene *Fgf9* and this negative feedback is thought to be an integral mechanism in establishing either an ovary or testis [112]. However, loss of *Fgf9* does not always cause sex reversal [113] and overexpression of *Wnt4*; therefore, suppression of *Fgf9,* in XY embryonic mouse gonads, affects the formation of testis vasculature and steroidogenesis but ultimately does not cause sex reversal [136]. The absence of *Wnt4* does not significantly change *Sox9* expression, suggesting that, when present, *Wnt4* is not suppressing the male pathway [10,112,123,137]. In contrast, loss of *Rspo1* does lead to upregulation of *Sox9,* suggesting the expression of *Rspo1* and its downstream action on *Ctnnb1* and *Wnt4* is critical for suppression of the male pathway. Similarly, *FoxL2* ablation in adult ovaries allows for upregulation of *Sox9* in the somatic cells [9], demonstrating an antagonistic relationship between these factors. 

β-catenin, which lies downstream of *Rspo1*, is suppressed by SRY in vitro in NTERA-2 clone D1 (NT2/D1) cells, a surrogate human Sertoli cell line [138]. SOX9 can similarly inhibit β-catenin in chondrocytes [139], but this has not been demonstrated in Sertoli cells. Conversely, β-catenin can also suppress transcription of *Sox9* in embryonic mouse gonads [4] and decrease the abundance of both SOX9 and AMH in NT2/D1 cells and embryonic mouse gonads [140]. Overall, this antagonistic relationship between SOX9 and β-catenin presents as a key regulator of gonad differentiation. 

More recently, mitogen-activated protein kinase (MAPK) pathways have been revealed to have a role in sex determination as mediators of the antagonistic relationship between SOX9 and β-catenin [141]. MAPK cascades are three-tiered, involving initial activation of a MAP kinase kinase kinase (MAP3K) by extracellular stimuli; activated MAP3Ks phosphorylate MAP kinase kinases (MAP2Ks), which in turn activate MAP kinases (MAPKs). The three classical MAPK pathways are extracellular signal-regulated protein kinases (ERK), c-Jun N-terminal kinases (JNK) and p38 MAP kinases. Two pathways, MAP3K4 and MAP3K1, have an interesting role in promoting or suppressing SOX9 or β-catenin, ultimately impacting the fate of the gonad [141]. 

MAP3K4 is responsible for a cascade of signalling leading to the initial expression of *Sry* in mouse gonads and mice deficient for *Map3k4* exhibit male-to-female sex reversal as a result of a decrease in *Sry* transcription [142]. Growth arrest and DNA damage-inducible protein γ (GADD45γ) is a binding factor of MAP3K4 [143] and facilitates the regulation of *Sry* transcription by the subsequent phosphorylation of p38 and GATA binding protein 4 (GATA4), allowing GATA4 and FOG2 to bind to the *Sry* promoter to upregulate its transcription [144,145]. Thus, the correct activation of MAP3K4 is required for the establishment of the testis pathway. In contrast, the loss of *Map3k1* in the mouse has little impact on testis development [146], suggesting it is not required for testis determination.

It is unknown what impact loss of *MAP3K4* has on testis development in humans, as mutations are likely embryonic lethal [141]; however, in human testis-derived cells, MAP3K4 can rescue the suppression of SOX9 caused by gain-of-function mutations in MAP3K1 [147], demonstrating it can promote the testis developmental pathway. The gain-of-function mutations in *MAP3K1* that result in suppression of SOX9 account for 13–20% of human gonadal dysgenesis cases [141]. These mutations lead to increased phosphorylation of p38 and ERK1/2 and increased binding of Ras homolog family member A (RHOA), Rho-associated coiled coil containing protein kinase (ROCK), FRAT regulator of Wnt signalling pathway 1 (FRAT1), and MAP3K4, as well as decreased binding of Rac family small GTPase 1 (RAC1) to MAP3K1. Together, these changes cause stabilisation of β-catenin and decreased expression of *SOX9*—thus, the activation of MAP3K1 can promote a shift to ovarian development [147,148,149]. This model demonstrates the complex role of MAP3K signalling and related factors in sex determination [141,147] (Figure 3). 

Research into the core pathways involved in mammalian gonad development has demonstrated that there are distinct genetic pathways required for the determination of either an ovary or testis. The expression and activity of these pathways is under the control of numerous factors, including the MAP3K1 and MAP3K4 cascades. While in normal circumstances these factors work in concert to reinforce the pre-existing gonad fate, extracellular changes, such as increased oestrogen signalling, can interfere with their activity. The antagonism between testis and ovary factors further reinforces the switch in somatic cell fate and altogether demonstrates that the fate of somatic cells in the gonad is plastic and that they can be influenced to form either a Sertoli or granulosa cell.

## 5. Targets of Oestrogen in the Gonad

Oestrogen signalling has critical roles in both male and female reproductive development. Oestrogen can promote a tilt in somatic cell fate from testis to ovary in many vertebrate species, even in the presence of genetic sex determination mechanisms [150,151]. Mammalian gonad development follows a robust genetic program and the initial determination of the ovary occurs in the absence of oestrogen; however, oestrogen is essential for the maintenance of granulosa cell fate and can have impacts on male reproduction when aberrant oestrogen signalling occurs, demonstrating the plasticity of these somatic cells. Thus, it is likely oestrogen has a conserved role in mammals in directing somatic cell fate away from a Sertoli cell and towards that of a granulosa cell.

Oestrogens are steroid hormones that require the binding of intracellular ERs to exert their widespread effects on cell function. Three types of ERs exist, the nuclear acting ERα (*ESR1*) and ERβ (*ESR2*), and the membrane bound G protein coupled receptor (GPER). ERα is the primary ER and can signal via numerous kinase pathways and transcriptional targets [152]. There are two distinct types of oestrogen signalling: genomic and non-genomic. Genomic oestrogen signalling is considered the classical pathway and involves either the direct binding of ligand-activated ERs to oestrogen response elements (EREs) in target DNA sequences [152], or the binding to transcription factors to form a complex that can then bind to DNA [153]. Non-genomic signalling involves ligand binding to plasma membrane-bound ERs that can rapidly activate kinase signalling, such as the MAPK pathway [154].

### 5.1. Non-Genomic Targets of Oestrogen in the Gonad

The non-genomic action of oestrogen has been well studied and both ERα and GPER have been implicated in the activation of numerous kinases [155]. There is a breadth of pathways that can be controlled by non-genomic oestrogen signalling but given that activation of ERK1/2 is able to promote ovarian fate by stabilising β-catenin [147], it presents as a potential target of oestrogen to suppress the male developmental program in this system. ERK1/2 is present in Sertoli cells, where it has a role in proliferation, among many other signalling pathways [156]. ERK1/2 can be activated by oestrogen in a non-genomic manner in breast cancer, bone, and neural cells [157,158,159,160]. Brief oestrogen treatment can also rapidly activate ERK1/2 in NT2/D1 cells to promote the cytoplasmic retention of SOX9 [161], demonstrating oestrogen can mediate SOX9 on both a non-genomic and genomic level. These results suggest oestrogen activates ERK1/2 in Sertoli cells to promote ovarian fate through stabilisation of β-catenin and inhibition of SOX9 (Figure 3). ERK1/2 is highly conserved [162]—thus, activation of ERK1/2 may be an ancestral mechanism through which oestrogen can direct somatic cell fate in vertebrates. Indeed, in the tammar wallaby, exposure of the developing gonad to oestrogen leads to increased expression of *MAP3K1* [163], which lies upstream of ERK1/2 and is a critical regulator of the gonad developmental programs. Mice lacking membrane-bound oestrogen receptors are protected from the impacts of exogenous oestrogens, such as DES [164], demonstrating this rapid response to oestrogen via membrane-bound ERs is likely the major way through which oestrogen impacts gonad development.

Oestrogen can similarly regulate the ovarian factor β-catenin through non-genomic mechanisms. In neurons [165], human colon cancer cells, and breast cancer cells [166], short term oestrogen treatment leads to the direct association of ERα with β-catenin to promote the activation of β-catenin. Furthermore, oestrogen treatment can dissociate β-catenin from the inhibitor glycogen synthase kinase 3β (GSK3β), eventually leading to decreased activity of GSK3β through activation of AKT serine/threonine kinase (AKT) signalling [167]. This suggests oestrogen can target GSK3β to reduce its inhibitory action on β-catenin. AKT signalling can also lead to direct activation of β-catenin via phosphorylation at serine (Ser)552 (Figure 3), increasing its transcriptional activity [168]. Oestrogen treatment rapidly activates AKT in breast cancer cells [169] and neurons [170] through the transmembrane oestrogen receptor GPER [171]—thus, it is possible AKT may also be activated in Sertoli cells exposed to oestrogen.

Protein kinase A (PKA) also promotes transcriptional activity of β-catenin via phosphorylation at Ser552, as well as Ser675 [172]. PKA activity is dependent on the levels of cyclic adenosine monophosphate (cAMP) [173] and can be induced following brief exposure to oestrogen in breast cancer and uterine cells [174]. PKA further promotes the activity of ERα via phosphorylation [175,176], suggesting it has a unique relationship in mediating ERα activity. p21 (RAC1) activated kinase 1 (PAK1) can also phosphorylate β-catenin at Ser675 (Figure 3) in colon cancer cells [177] and can be activated by oestrogen in breast cancer cells [178], while its transcription is also oestrogen responsive [179].

Altogether, the activation of ERK1/2, AKT, PKA, and PAK1 present as potential targets of oestrogen to promote ovarian fate in Sertoli cells (Figure 3); however, it is difficult to predict how these kinases may respond in a different cell type and what impacts their activation would have on other aspects of the cell. The findings that oestrogen can rapidly activate ERK1/2 to suppress SOX9 [161] demonstrates how essential assessing the effects of oestrogen on non-genomic targets is, as this type of signalling often establishes the changes required for genomic signalling to occur. Furthermore, these signalling pathways are critical for spermatogenesis and have been linked to male infertility [180], further supporting the impacts of exogenous oestrogen on non-genomic pathways and declining male reproductive health.

### 5.2. Genomic Targets of Oestrogen in the Gonad

Oestrogen can directly inhibit transcription of *SOX9* in the red-eared slider turtle (*Trachemys scripta*) [181], chicken [182], and the broad-snouted caiman (*Caiman latirostris*) [183]. In mammals, the best example of the ability of oestrogen to impact gonad somatic cell fate on a genomic level comes from research in marsupials. In the tammar wallaby, oestrogen exposure of XY embryonic gonads for 5 days does not decrease transcription of *SOX9*; however, it does lead to the cytoplasmic retention of SOX9 protein [11,12] (Figure 4). This suppression of SOX9 activity causes sex reversal and transdifferentiation of Sertoli cells to granulosa-like cells. These granulosa-like cells exhibit upregulation of ovarian markers *FOXL2* and *WNT4* and reduced expression of *SRY* and *AMH* [11,12] —thus, oestrogen is able to tilt the balance from testis to ovarian fate in marsupial gonads.

Exogenous oestrogen similarly affects SOX9 subcellular localisation in human testis-derived NT2/D1 cells, leading to suppression of SOX9 target genes *FGF9*, *PTGDS*, and *AMH* and activation of *WNT4* and *FOXL2* [11] (Figure 4). These results demonstrate that oestrogen can influence the key gonadal factors involved in determining somatic cell fate of the human gonad. The cytoplasmic retention of SOX9 by oestrogen presents as a mechanism through which oestrogenic EDCs can impact Sertoli cells and testis development and function. In humans, the requirement for SOX9 nuclear localisation to drive testis differentiation is well established, and mutations affecting SOX9 import are associated with DSDs [184]. This mechanism may contribute to infertility in adult males with elevated oestrogen levels [52]. These findings are important for understanding how disruption to ovarian steroidogenesis may impact granulosa cell fate and ovarian maintenance. A loss of oestrogen signalling—such as in POI and PCOS—could lead to an increase in SOX9 activity and disruption of granulosa cell fate.

There is further evidence to suggest oestrogen can impact the transcriptional profile of gonad somatic cells in mice. In adult mouse ovaries, *Sox9* transcription can be suppressed by the combined action of activated ERα and *FoxL2* on the SOX9 enhancer TESCO, and this is an important step in maintaining granulosa cell fate [9]. FOXL2 can also directly activate *Esr2* (ERβ) transcription to suppress *Sox9* transcription and promote granulosa cell fate in adult mouse ovaries [132].

The expression of some downstream targets of SOX9 are oestrogen responsive—*FGF9* and its receptor *FGFR1* have oestrogen response elements [185,186] and their transcription can be directly targeted by oestrogen, while *AMH* undergoes differential regulation in response to oestrogen depending on cell type. In mature granulosa cells, ERα activation upregulates *Amh* [187] and its expression is essential for folliculogenesis in mice and humans [188,189]. In contrast, exposure of male rats to oestrogenic endocrine disruptors causes a decrease in *Amh* mRNA levels [190,191], alongside disruption in testis function. This effect may be due to suppression of *Sox9*; however, these results demonstrate *Amh* expression is a good indicator for disruptions to testis development. Another downstream target of Sox9, *Ptgds*, can similarly be inhibited by increased oestrogen signalling in mouse Leydig cells [192] and hypothalamus [193]. Together, these data demonstrate that oestrogen can target key testis pathway genes, however, inhibition of SOX9 presents as the most detrimental to testis development given it is the orchestrator for expression of the essential testis genes.

In contrast, there is less evidence to demonstrate that oestrogen can promote expression of ovarian factors. As mentioned above, *FoxL2* works in conjunction with oestrogen receptors to inhibit *Sox9* expression in the adult mouse and its expression is significantly increased following oestrogen treatment in wallaby and NT2/D1 cells. *FoxL2* KO mice show a decrease in expression of aromatase [194], further suggesting a link between oestrogen signalling and *FoxL2* expression. Long term oestrogen treatment can increase *Ctnnb1* transcription in mouse prostate [195] and uterus [196], and can reduce the transcriptional activity of AXIN1 (a member of the β-catenin degradation complex) in breast cancer cells, overall suggesting oestrogen can promote stabilisation of β-catenin [197]. *Wnt4* is activated in rat neurons following oestrogen exposure [198] but this has not been examined in gonads. There is little evidence that exogenous oestrogen can activate *RSPO1* or *FST* expression in humans and mouse and these genes did not respond to oestrogen treatment in the tammar wallaby [11,12]. However, it is possible β-catenin activation by oestrogen could lead to their upregulation in humans and mouse. Overall, it is highly likely some of these genes are responsive to oestrogen, as their continued expression is required to maintain granulosa cell fate and therefore to support the production of oestrogen.

## 6. Conclusions

Defining the mechanisms through which oestrogenic EDCs impact the gonads is essential for understanding the aetiology of DSDs and how these chemicals can impact reproductive development. The rapid decline in human reproductive health has been unequivocally linked to increasing exposure to oestrogenic chemicals in our environment. Here, we have described the known pathways through which gonadal fate decisions are made and the many ways these pathways can be impacted by exposure to oestrogenic chemicals. It is now clear that exogenous oestrogen can target both non-genomic and genomic pathways in the somatic cells of the gonad to affect cell fate decisions and their long-term maintenance. In particular, oestrogen impacts the somatic cells through alterations to MAPK signalling and the subcellular localisation of SOX9, leading to suppression of testis genes and activation of ovarian genes. These effects ultimately disrupt both the development and function of the gonad. Clearly any EDC that alters oestrogen signalling will profoundly impact gonad development and function.

## Figures and Tables

**Figure 1 ijms-21-08377-f001:**
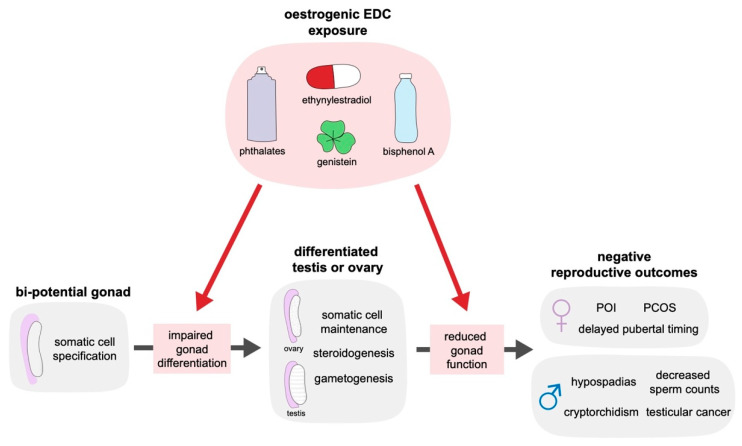
Exposure to oestrogenic endocrine disrupting chemicals, such as genistein (present in soy and subterranean clover), bisphenol A (a plasticiser), ethynylestradiol (a synthetic oestrogen present in the contraceptive pill), or phthalates (plasticisers, present in some cosmetics), can disrupt gonad development and function, leading to negative reproductive outcomes. Exposure of the bi-potential gonad as it undergoes differentiation into either an ovary or testis can disrupt somatic cell specification, an important step in establishing gonad fate. Exposure of the differentiated gonad can impact steroidogenesis, gametogenesis and the ongoing maintenance of somatic cell fate, disrupting the continuing function of the gonad. Either periods of exposure can contribute to the development of premature ovarian insufficiency (POI), polycystic ovary syndrome (PCOS), or cause delayed pubertal timing in females. In males, such exposures have been linked to testicular dysgenesis syndrome, comprising of hypospadias, cryptorchidism, decreased sperm counts, and testicular cancer.

**Figure 2 ijms-21-08377-f002:**
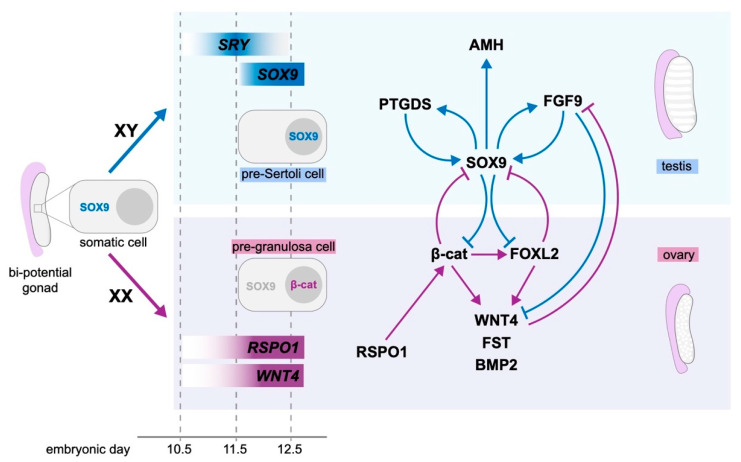
Key genetic pathways involved in gonad differentiation in mouse. SRY-box transcription factor 9 (SOX9) is present in the cytoplasm of the somatic cells of both XX and XY bi-potential gonads. In XY mouse gonads, expression of sex-determining region Y (*Sry*) reaches a peak at embryonic day (E) 11.5 and triggers the nuclear translocation of SOX9, where it promotes expression of prostaglandin D synthase (*Ptgds*), fibroblast growth factor 9 (*Fgf9*), anti-Müllerian hormone (*Amh*), and itself, together contributing to the differentiation of a testis. In XX gonads, in the absence of *Sry*, SOX9 remains cytoplasmic. R-spondin 1 (*Rspo1*) and Wnt family member 4 (*Wnt4*) are expressed specifically from E12.5, and β-catenin is stabilised in the nucleus, while the cytoplasmic pool of SOX9 disappears. The activity of these ovary-specific genes triggers expression of other genes forkhead box L2 (*FoxL2*), follistatin (*Fst*), and bone morphogenetic protein 2 (*Bmp2*) to promote ovarian differentiation. SOX9 further promotes testis development in males by inhibiting β-catenin and FOXL2 to ensure ovarian development is suppressed. Conversely, β-catenin and FOXL2 inhibit SOX9 to promote ovarian differentiation. WNT4 and FGF9 also exhibit antagonism.

**Figure 3 ijms-21-08377-f003:**
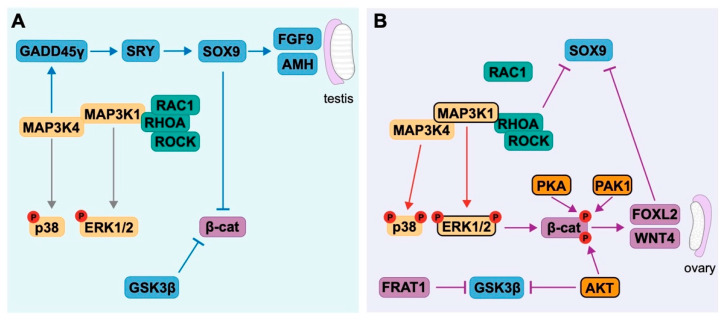
The influence of mitogen-activated protein kinase kinase kinase (MAP3K) cascades on gonad differentiation and potential non-genomic targets of oestrogen. Pro-testis factors are shown in blue and pro-ovary factors in purple. (**A**) In testis development, MAP3K4 together with growth arrest and DNA damage-inducible protein γ (GADD45γ) promotes expression of *SRY*, which in turn promotes *SOX9* and testis development. Together with SOX9, glycogen synthase kinase 3β (GSK3β) inhibits β-catenin activity, blocking ovarian development. (**B**) In ovarian development, activation of MAP3K1 and MAP3K4 and increased binding of MAP3K1 to RHOA/ROCK promotes phosphorylation of p38 and ERK1/2. Activated ERK1/2 and FRAT1-mediated inhibition of GSK3β promotes stabilisation of β-catenin and expression of ovarian genes *FOXL2* and *WNT4*. Phosphorylation of β-catenin by AKT serine/threonine kinase (AKT), protein kinase A (PKA), and p21 (RAC1) activated kinase 1 (PAK1) promotes its activity and AKT activation reduces the activity of GSK3β. SOX9 is inhibited through the activity of β-catenin, FOXL2 and RHOA. Potential non-genomic targets of exogenous oestrogen that could promote granulosa cell fate are indicated by a black border. Adapted from Ostrer 2014 [141].

**Figure 4 ijms-21-08377-f004:**
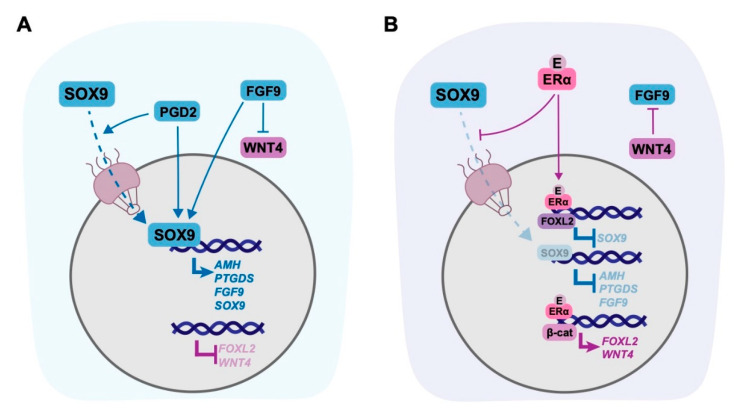
Model for the regulation of gonadal genes and SOX9 subcellular localisation by oestrogen in Sertoli cells. (**A**) In a normal Sertoli cell SOX9 increases expression of itself and its downstream targets *AMH*, *FGF9*, and *PTGDS* by translocating from the cytoplasm to the nucleus. PGD2 facilitates the nuclear entry of SOX9, while FGF9 inhibits WNT4 and there is no expression of *FOXL2*. (**B**) Exogenous oestrogen (E) blocks SOX9 nuclear entry, preventing activation of SOX9 downstream targets. Activated oestrogen receptors (ERα) and FOXL2 repress SOX9 transcription and, together with β-catenin, promote expression of *WNT4* and *FOXL2*. WNT4 subsequently inhibits FGF9.

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
