# Peer review of "Exogenous Oestrogen Impacts Cell Fate Decision in the Developing Gonads: A Potential Cause of Declining Human Reproductive Health"

_ijms, 2020, doi:10.3390/ijms21218377_

Round 1

Reviewer 1 Report

General comments:

The review submitted by Stewart and colleagues deals with the potential for endogenous ‘estrogenic compounds’ to interfere with mammalian gonad development and, by so doing, cause reproductive disorders. It is a very thorough review with a substantial reference list. It is obvious that the authors are experts in the area of gonadal sex determination and differentiation. They have also covered the fields of ‘chemical endocrine disruption’/toxicology well, but with some smaller errors/omissions. If anything, some of the text on the genetic control of gonadal sex determination may be a bit excessive with regard to the focus of the review, but it nevertheless paint a more complete picture for those readers not experts in the field.

Overall, it is a very well written review with some interesting, and important, observations. After all, the ‘oestrogen hypothesis’ of Testicular Dysgenesis Syndrome, proposed by Sharpe & Skakkebaek back in the early 1990s, has later been the source of much scientific debate. Sharpe and colleagues have taken the view that oestrogenic EDCs don’t play any role in TDS in humans based on the apparent lack of ERa/ESR1 in human fetal testis, whereas Skakkebaek and others have maintained the view that this oestrogenic EDCs still play a role (See for instance Sharpe 2003, Int J Androl 26:2-15 and Mitchell et al 2013, PLoS One 8:e61726). I am not suggesting bringing up this debate in the present review, but discussion around rodent versus human fetal gonads and potential difference in expression of the two isoform should be addressed (see point X below)?

My overall assessment is that this is a timely and important review that would be a valuable addition to the details around estrogenic EDCs and reproductive disorders. I have no major issues with the paper, but a few minor comments/concerns I feel should be adressed, as detailed below.

Specific comments:

  1. Definition of EDCs on lines 61-62 is incomplete (thus, incorrect). It is true that La Merrill et al mention NRs as a modality, but also several others. In fact, the canonical modalities to classify a substance as an EDC is EATS (estrogenic, androgenic, thyroid and steroidogenesis), but there are also several other less studied modalities not necessarily involving NRs. And for reproductive disorders, steroidogenesis is a frequently encountered modality which does not specify direct effects on NRs (the authors are themselves talking about S modality on e.g. lines 139-141). I suggest to instead use the more standard definition by WHO: “an exogenous substance or mixture that alters function(s) of the endocrine system and consequently causes adverse health effects in an intact organism, or its progeny, or (sub) populations” - IPCS, Global assessment of the state-of-the-science of endocrine disruptors, in International Program on Chemical Safety, WHO/PCS/EDC/02.2, T. Damstra, et al., Editors. 2002, WHO.
  2. Line 67; when listing potential oestrogenic EDCs, why are phthalates included? Phthalates belong to one of the most studied of all EDCs and are overwhelmingly considered anti-androgenic. I suggest removing reference to phthalates in this context, also removing them from Fig 1. Alternatively, the authors could consider adding parabens, for which there are several studies suggestion estrogenic mode of action in male reproduction. To find relevant publications, refer to e.g. extensive ED assessment review of butylparaben by Boberg et al 2020, Environ Int 144:105996.
  3. Lines 72-73; it is correct as stated that mammalian gonads express ERs throughout development, but according to Mitchell et al 2013 (see above) only ERb/ESR2 is expressed in human fetal testis and hence, since relevant action should be mediated by ERa, you would not see an effect in humans as you do in rodents. This topic must be adressed in some way, either the authors agree or disagree with this notion.
  4. Lines78-80, the citations in the end of this compound sentence are incorrectly placed. They should become before the last clause, otherwise it sound like they are citations to back up the statement that AGD is a biomarker for compromised androgen signaling, which they are not. I suggest reformatting to: “In mice, exposure to the oestrogenic endocrine disruptor diethylstilbestrol (DES) in utero leads to increased rates of hypospadias and reduced anogenital distance (36,37), a maker of androgen output during development (ref: e.g. Schwartz et al 2019, Arch Toxicol 93:253-272; Thankamoney et al 2016, Andrology 4:616-625)
  5. Lines 107-108, this again is true, but seems downplayed with only reference to #47. Authors could consider referring to the Ovarian Dysgenesis Syndrome (since TDS is given great weight earlier): Johansson et al 2017, Nat Rev Endocrinol 13:400-414.
  6. Lines 172-203; these are very interesting studies indeed. But should the authors also consider the effects Tamoxifen might have on gonad differentiation? Tamoxifen being an ER blocker is of course the ‘opposite’ of estrogenic, but still may help illuminate the role for ER action in gonad differentiation. It is a bit of a contentious issue, since tamoxifen has been used as an inducer of conditional Kos in the field for very long time (!), but still it cannot be swept under the carpet that in may have effects on gonad determination. See Patel et al 2017, Sci Rep 7:8991
  7. Section 5.1 (line 422). Are the authors aware of the study Nanjappa et al 2019, Biol Reprod 101:392-404, where they show in mice without membrane bound ERa are protected from estrogenic EDCs? If this paper holds true, it could have many implications for the mechanisms/modalities discussed in this review? I think the findings of Nanjappa must feature in this review somehow and given some considerations.
  8. Conclusion, line 532. “Defining the mechanisms through which these chemicals impact….” You should be more precise and specify what you mean by ‘these chemicals’.
  9. Line 52, to me it sounds more logical to write ’development and functioning’ in this order rather than opposite.

Reviewer 2 Report

I commend the authors for describing this critical and timely issue. The paper is interesting and well written, and provide insights into the pathways by which endocrine disruptors with oestrogen-like activity can influence somatic cell fate to disrupt gonad development and function.

I have only minor comments.

Line 174. What are “the other vertebrate species mentioned”? Did you mean “other mammals such as mice and rats”?

Line 195. Please provide the full name of GSD.

Line 199. What did the authors mean with “the indifferent gonad”? Probably “indifferentiated”?

Line 233. Please change to “in the absence of Sry”…

Line 235. “Exogenous expression of Sox9” in the sense of “Expression of Sox9 triggered by exogenous substances”?

Author Response

I commend the authors for describing this critical and timely issue. The paper is interesting and well written, and provide insights into the pathways by which endocrine disruptors with oestrogen-like activity can influence somatic cell fate to disrupt gonad development and function.

I have only minor comments.

Line 174. What are “the other vertebrate species mentioned”? Did you mean “other mammals such as mice and rats”?

We have changed to this to “non-mammalian vertebrates”.

Line 195. Please provide the full name of GSD.

Thank you to the reviewer for noticing this. It has been amended.

Line 199. What did the authors mean with “the indifferent gonad”? Probably “indifferentiated”?

‘Indifferent gonad’ is the correct term for this structure in this field and is how it is described in all other papers. 

Line 233. Please change to “in the absence of Sry”…

We have removed “in XX gonads”

Line 235. “Exogenous expression of Sox9” in the sense of “Expression of Sox9 triggered by exogenous substances”?

We have altered this to “ectopic”